# Bridging the Chemical Profile and Biomedical Effects of *Scutellaria edelbergii* Essential Oils

**DOI:** 10.3390/antiox11091723

**Published:** 2022-08-30

**Authors:** Muddaser Shah, Shabana Bibi, Zul Kamal, Jamal Nasser Al-Sabahi, Tanveer Alam, Obaid Ullah, Waheed Murad, Najeeb Ur Rehman, Ahmed Al-Harrasi

**Affiliations:** 1Department of Botany, Abdul Wali Khan University Mardan, Mardan 23200, Pakistan; 2Natural and Medical Sciences Research Center, University of Nizwa, Birkat Al-Mauz, P.O. Box 33, Nizwa 616, Oman; 3Department of Biosciences, Shifa Tameer-e-Millat University, Islamabad 44000, Pakistan; 4Yunnan Herbal Laboratory, College of Ecology and Environmental Sciences, Yunnan University, Kunming 650091, China; 5Department of Pharmacy, Shaheed Benazir Bhutto University, Upper Dir 18000, Pakistan; 6Central Instrument Laboratory, College of Agriculture and Marine Sciences, Sultan Qaboos University, Muscat 123, Oman; 7Department of Chemistry, University of Chakdara, Chakdara 18800, Pakistan

**Keywords:** *S. edelbergii*, essential oils, GC-MS analysis, in vitro and in vivo biological activities

## Abstract

The present study explored chemical constituents of *Scutellaria edelbergii* essential oils (SEEO) for the first time, extracted through hydro-distillation, and screened them against the microbes and free radicals scavenging effect, pain-relieving, and anti-inflammatory potential employing standard techniques. The SEEO ingredients were noticed via Gas Chromatography-Mass-Spectrometry (GC-MS) analysis and presented fifty-two bioactive compounds contributed (89.52%) with dominant volatile constituent; 3-oxomanoyl oxide (10.09%), 24-norursa-3,12-diene (8.05%), and methyl 7-abieten-18-oate (7.02%). The MTT assay via 96 well-plate and agar-well diffusion techniques against various microbes was determined for minimum inhibitory concentration (MIC), minimum bactericidal concentration (MBC), IC_50_, and zone of inhibitions (ZOIs). The SEEO indicated considerable antimicrobial significance against tested bacterial strains viz. *Escherichia coli*, *Pseudomonas aeruginosa*, *Klebsiella pneumoniae*, and *Enterococcus faecalis* and the fungal strains *Fusarium oxysporum* and *Candida albicans*. The free radicals scavenging potential was noticed to be significant in 1,1-Diphenyl-2-picryl-hydrazyl (DPPH) as compared to 2,2′-azino-bis-3-ethylbenzotiazolin-6-sulfonic acid (ABTS) assays with IC_50_ = 125.0 ± 0.19 µg/mL and IC_50_ = 153.0 ± 0.31 µg/mL correspondingly; similarly, the antioxidant standard in the DPPH assay was found efficient as compared to ABTS assay. The SEEO also offered an appreciable analgesic significance and presented 54.71% in comparison with standard aspirin, 64.49% reduction in writhes, and an anti-inflammatory potential of 64.13%, as compared to the standard diclofenac sodium inhibition of 71.72%. The SEEO contain bioactive volatile ingredients with antimicrobial, free radical scavenging, pain, and inflammation relieving potentials. Computational analysis validated the anti-inflammatory potential of selected hit “methyl 7-abieten-18-oate” as a COX-2 enzyme inhibitor. Docking results were very good in terms of docked score (−7.8704 kcal/mol) and binding interactions with the functional residues; furthermore, MD simulation for 100 ns has presented a correlation with docking results with minor fluctuations. In silico, ADMET characteristics supported that methyl 7-abieten-18-oate could be recommended for further investigations in clinical tests and could prove its medicinal status as an anti-inflammatory drug.

## 1. Introduction

Medicinal herbs are the most prevalent form of conventional therapies used for maintaining human health, in addition to inhibiting or curing physical and mental diseases [1,2]. Plants produce diverse chemical ingredients, such as essential oils (EOs), which have multiple properties; for example, they can resist microbes [3], scavenge free radicals [4], cure inflammation [5], and relieve pain [6]. The EOs are complex concoctions with an aroma accumulated in the special plant cells and are obtained from the aromatic plants by hydro wave, microwave, or steam distillation.

They are highly efficacious and are economically important due to lower toxic effects as compared to the available commercial drugs. Thus, the EOs have been reported and tested as alternative remedies and are particularly important antimicrobial agents [7]. The widespread and irrational use of the currently used antibiotics has resulted in microbial mutation and antimicrobial resistance, thus lowering their therapeutic efficacy. This, in turn, increases the demand for plant-based constituents as alternatively safe and effective antimicrobial resources [8,9]. The EOs offered promising potential against the microbes as well as presented the significant capability to scavenge the free radicals with fewer side effects [10].

Oxidation is a biochemical process involving the transfer of an electron from a rich to a deficient unit. The deficient electron molecule is named an oxidizing agent [11]. The agents required for neutralizing the effects of the oxidizing agents are termed antioxidant agents, thus protecting our body from cellular damage because of oxidation effects [12]. The damaging effects of reactive oxygen species (ROS) are well-adjusted by the antioxidant activity of plant-based natural products [13].

These properties are interconnected and lead toward the control of inflammatory objectives. Hence, this chemical constituent proved itself as an essential mediator to incite or maintain the inflammatory condition and balance the human body by scavenging free radicals [10]. It is also reported that anti-oxidation, inflammation, and pains are interconnected objects that affect other mechanisms in the body; this is because oxidative stress can influence a variety of transcriptional factors and hence could the differential expression of important genes and gene elements affecting inflammatory mechanism [14]. Pain is a distasteful sensation triggered through internal or external stimuli, and it can also influence the sensory experience related to tissue damage [15]. The sensation of pain and its response to analgesics have an integrated relationship that involves many biochemical channels that are controlled by significant hereditary factors and may modify the pain stimuli or hinder the response to analgesics [12]. Moreover, there is a terrific deal of inter-individual variability in pain perception as well as the dosage required to relieve pain, and EOs have the potential to cope with and overcome pain [16]. Inflammation is the human body’s defensive mechanism to combat chemical, physical, or biological hostility. Thus, it is the basic need to annihilate the detrimental agent and diminish its toxic effect by reducing its propagation [12]. Furthermore, the inappropriate usage of EOs obtained from plants belonging to the family Rutaceae might cause adversative properties in human beings, irritation of the skin, inflammation, headache, and nausea. Vigilance is usually needed if EOs is to be taken internally or used on foodstuff for the reason of the probable cancer-causing consequences among them [17]. With the use of a non-recommended dosage, EOs might cause functional harm to organs such as the stomach and liver in animals and, almost certainly, in humans [18].

The currently used anti-inflammatory and antioxidant drugs contain steroidal or non-steroidal anti-inflammatory constituents, and prolonged use is associated with adverse effects such as gastrointestinal intolerance, depression of bone marrow, and retention of salt and water [19]. Natural remedies, on the other hand, are an alternative source with a higher therapeutic efficacy rate and fewer side effects. Hence, the essential oils bearing herbs and their products can overcome inflammatory-related complications [20]. Lamiaceae is a cosmopolitan family and contains around 252 genera and 6700 taxa [21]. Mostly the species of Lamiaceae contain terpenes and many other bioactive constituents, predominantly occurring in the epidermal glands of flowers, leaves, and stems [22]. The genus *Scutellaria* L. comprises 360 plant species, distributed, and practiced as healing agents in conventional medicines in China, Korea, North America, and Pakistan [1,2,23]. Currently, more than 300 bioactive ingredients, including terpenes, flavonoids, flavones, glycosides, and terpenoids, have been isolated from *Scutellaria* [23,24], which have multiple health benefits; thus, *Scutellaria* species are well-known for their diverse promising medicinal purposes and offered significance resistance to the microbes, also act as an antioxidant, anti-inflammatory, and analgesic agent [25]. *Scutellaria edelbergii* Rech. F. (Lamiaceae) is a perennial herb, spreading through its hard woody rootstock, with slender stems and a procumbent or softly ascending, round-quadrangular, much branched ovate or acute margins leaves. The flowers are observed subtended, with a small calyx with purple scutellum, become puffy during fruit, yellow petals or occasionally blue-violet, lower lip darker, spreading erect or erect, densely glandular pilose. The flowering period is April–July, found between 1660 and 2200 m [26].

*S. edelbergii* is locally named panra and is used to cure inflammation and relieve pain, make green tea (kawa), and purify the blood. It is distributed over North America, Europe, East Asia, and Pakistan. However, in Pakistan, *S. edelbergii* is found in the mountain trails of Swat, Kalam, and Chitral [1], with moist loamy soil habitats. The EOs composition of the plant is influenced by climatic, topographic, edaphic, harvesting time, quality, and availability of water. The EOs of the genus *Scutellaria* mainly contain α-pinene, β-pinene, thymol, myrcene, linalool, sabinene, and γ-eudesmole, which are well known for their biomedical significance as reported by [27,28,29,30]. The crude extract and subfractions of the selected plant *S. edelbergii*, as well as its n-hexane extracted fatty acid esters, were found as antimicrobial, antioxidant, antidiabetic, analgesic, and anti-inflammatory agents [31]. However, currently, no scientific data are available for the therapeutic applications of *S. edelbergii* essential oil. Thus, the present exploration was designed to describe the SEEO constituents and screened them for their antimicrobial, antioxidant, analgesic, and anti-inflammatory effects and further validated using a computational approach.

## 2. Materials and Methods

### 2.1. Plant Material

The fresh *S. edelbergii* plant species were gathered from mountain tails in the Kalam region (Ushu and Mataltan), District Swat, at random intervals (April–June 2019), during the flowering season. After collection, *S. edelbergii* plant species were shifted to the research laboratory and placed in the open air to dry. The air-dried whole plant (8.7 kg) was pressed and preserved for identification and placed at the herbarium at the Department of Botany, Abdul Wali Khan University, Mardan (AWKUM/Herb/2234).

### 2.2. Essential Oils Extraction

The understudy plant species were chopped using an electric blender after they were completely dried. The obtained powder (2.0 kg) was weighed, and the essential oils (1.6 g) were extracted using a Clevenger device via hydro-distillation for 6 h, three times, until no more essential oils could be extracted. The EOs (0.08%) were measured after being collected off the top of the hydrosol in a glass bottle and were passed through anhydrous sodium sulfate to remove the moisture and placed in the refrigerator until further use [32].

### 2.3. GC-MS Analysis and Compounds Identification

The bioactive ingredients in the SEEO were determined using gas chromatography-mass spectrometry (GC-MS) analysis using a Perkin-Elmer-Clarus (PEC) 600 GC device, coupled with Rtx5MS, with a capillary column (30 m × 0.25 µm) at a maintained temperature of 260 °C attached with PEC 600 MS. The ultra-pure helium (99.99%) was used as a carrier gas at a constant flow rate of 1.0 mL/min. The injection, transfer line, and ion source temperatures were 260 °C, 270 °C, and 280 °C, respectively. The ionizing energy was noted as 70 eV. The electron multiplier voltage was operated from auto-tune. The full-scan mass spectra were obtained at 45–550 a.m.u. scan range. The tested sample at 1 µL quantity was loaded with a specified split ratio of 10:1. The oven temperature was maintained at 60 °C for a minute, while the temperature from 4 °C/min up to 260 °C was maintained for 4 min. The process was completed in 50 min [31]. The chemical constituents were identified through their corresponding chromatograms peaks obtained for each oil via GC-MS analysis in terms of retention indices (RI) compared with standards and the spectral mass data of each chromatogram through the National Institute of Standard Technology NIST-14 (2011 Ver. 2.3) [33] and were further authenticated by using the available literature [32].

### 2.4. Antimicrobial Screening

The EOs were examined for their antimicrobial screenings from low to high dosages. OD_600nm_ through the 96-well-plate method was used to determine MIC, whereas its agar-plate method was used to calculate MBC, as ZOIs were carried out by employing the agar-well diffusion technique [31]. The clinical isolates microbial strains were identified and authenticated by Chairman Dr. Hazir Rahman, Department of Microbiology AWKUM, Mardan.

#### 2.4.1. Antibacterial Assay

Fresh bacterial strains inoculum of *K. pneumoniae*, *P. aeruginosa**, E. coli,* and *E. faecalis* from a single colony were transferred to a sterile nutrient broth media and kept overnight at 37 °C/220 rpm. The MIC at OD_600nm_ (~0.5 McFarland standards) of *S. edelbergii* essential oil were determined for 1, 5, 10, 15, 25, 50, 100, 150, 200, and 500 µL two-fold serial dilution in sterile broth concentrations. Similarly, 20 µL of respective bacterial inoculum (1.5 × 10^6^ CFU) were added to 100 µL of each concentration in triplicate wells and incubated for 18-24 h at 37 °C ± 0.5. Then, 50 µL of 3-(4,5-dimethylthiazol-2-yl)-2,5-diphenyltetrazolium bromide (0.2 mg/mL) MTT were added, and the plate was incubated at 37 °C for 45 min, along with an appropriate sterile broth, run as negative control while the bacterial suspension was comprised as a positive control. The absorbance was measured at 570 nm, and IC_50_ = OD of positive control-OD of test sample/OD of positive control x100 were calculated. Similarly, MBC for all these tested samples was confirmed by the agar plate spreading method (100 µL from MIC results of all concentrations were coated on TSA plates and cultured overnight, zero growth shows MBC. For scientific validity and authenticity, the entire data were taken in triplicates and listed as (Mean ± SEM).

The above available bacterial strains at a concentration of 1.5 × 10^8^ CFU/mL of the bacterial cell density (BCD) were spread over the solidified media using a wire loop. In these Petri dishes, four wells at the same distance of size 3 mm were made. The essential oils at 50 µg⁄mL and 100 µg⁄mL was injected through micropipette into the 1st and 2nd wells, while the negative control (DMSO) and levofloxacin as standard was employed in the 3rd and 4th well, respectively, and were incubated at ±37 °C for 24 h and the obtained ZOI was measured in mm.

#### 2.4.2. Antifungal Assay

The same procedures as those used in the antibacterial assay were used for antifungal activity assays on the SEEO. The *C. albicans* and *F. oxysporum* strains were sub-cultured on a fresh potato dextrose agar (PDA) plate for 24 h before antifungal assays. The same dose and same concentrations were used for SEEO to determine MIC, minimal fungicidal concentrations (MFC), and ZOIs [31]. The DMSO was run as negative control, while clotrimazole was used as a positive control accordingly. Then, during the incubation, the zone of inhibition was measured in mm. For scientific validity and authenticity, the entire data were taken in triplicates and listed as (Mean ± SEM).

### 2.5. Antioxidant Activity

#### 2.5.1. DPPH Assay

The SEEO were screened for their free radicals scavenging effect using the DPPH assay [31,34] with a slight variation. The DPPH 3 mg was dissolved in 100 mL distilled methanol. The homogenized solution was placed in the dark for 30 min to generate free radicals for investigating the antioxidant activity of the EO. The samples were tested at different concentrations of 1000, 500, 250, 125, and 62.5 µg/mL. Next, 2 mL from each sample was mixed with 2 mL of the already prepared DPPH solution and placed for incubation in the dark for 25 min. The absorbance of the test samples was then determined at 517 nm using UV/Vis spectrophotometry. Ascorbic acid was employed as standard. The antioxidant potential of the test samples was determined using the following Equation (1).
% Scavenging activity = A − B ⁄A × 100(1)
where (A) absorbance of the control and (B) is the absorbance of the tested samples (standard and essential oil).

#### 2.5.2. ABTS Assay

The free radicals scavenging significance of the tested samples was conducted using ABTS assay. About 383 mg of ABTS and 66.2 mg of K_2_S_2_O_8_ were separately dissolved in 100 mL analytical grade methanol and then combined. Next, 2 mL from the ABTS solution was incubated with 2 mL of test samples for 25 min using similar concentrations as described in the earlier DPPH assay. Furthermore, the absorbance of the EOs and ascorbic acid was determined at 746 nm using UV/V is spectrophotometry. The free radical scavenging effect was estimated using Equation (1).

### 2.6. Approval of Experimental Animals

Healthy Swiss mice of (24–30 g) were obtained from Veterinary Research Institute (VRI), Peshawar, and were accommodated in cages in AWKUM animal house under a controlled temperature of 20 °C for 6.5 weeks. The required materials (rodent pellets, foodstuff, and water) were given under cleaned conditions following ARRIVE guidelines.

### 2.7. Analgesic Activity

The EOs of the understudy plant was tested for relieving pain using an acetic acid-induced writhing assay as stated by [31] with slight modification. Moreover, swiss albino mice were used and divided into 5 groups (*n* = 6). The EOs, control, and standard were injected into the mice through intraperitoneal muscle with a sterilized syringe. All the mice groups were pretreated with 1 mL acetic acid (0.7%) at a concentration of 5 mL/kg body weight (BW) and then after 45 min. Next, the experimental animal of group 1 indicated as normal control, was treated with 1 mL of normal saline, while the animals in Group 2 were infused with 1 mL aspirin as a standard. Furthermore, the SEEO was administrated to the remaining swiss albino mice in groups 3, 4, and 5 at doses of 25, 50, and 100 mg/kg BW doses, respectively.

Writhes numbers were counted for determining the analgesic effect of the tested samples in comparison with normal saline and standard for 10 min. The results obtained were expressed in % inhibition using Equation (2).
(2)% Inhibition=A−BA×100
(A) is the writhes inducer (acetic acid); while (B) is the tested samples significance (EO, standard, and control).

However, in the anti-inflammatory assay, (A) in the equation indicates the paw edema induced through carrageenan.

### 2.8. Anti-Inflammatory Activity

The efficacy of SEEO to treat inflammation was tested using carrageenan-induced paw edema in Swiss albino mice, as stated by [31], with slight modification. The mice were grouped as mentioned in the analgesic activity: The inflammation was induced by injecting 1 mL (1%) carrageenan into all six groups of the swiss albino mice [35]. After 30 min, the 1st group was given 1 mL of normal saline, while group 2 was given 1 mL of diclofenac Na (50 mg/kg) using a sterile syringe. The SEEO at 25, 50, and 100 mg/kg body weight doses were injected into groups 3, 4, and 5, respectively, following the safety measures. The anti-inflammatory activity of tested samples was observed by measuring the paw diameter of the experimental animals right after each 1st, 2nd, and 3rd hour, respectively. In addition, the resulting data were expressed as % inhibition and calculated using Equation (2).

### 2.9. Computational Analysis

#### 2.9.1. Construction of Chemical Compounds Database

The literature is available to highlight the importance of identifying promising anti-inflammatory medicines and already available synthetic drugs on the market that are not so healthy because of certain adverse side effects [36]. Computer-assisted screening applications highly support the identification of novel drugs for different diseases [37,38]. An integrated computer-assisted scheme was employed using the database of 52 compounds to identify novel drugs to combat anti-inflammatory diseases. The structure of each compound was drawn by using the ChemDraw software [39], and information on each structure was crosschecked from the PubChem database [40] to reduce the chance of ambiguity and saved in SDF format for further analysis.

#### 2.9.2. Selection of Target Protein

It is highly significant to understand the disease mechanism and then select the appropriate protein structure to initiate the drug design pipeline, and it could explain the important parameters necessary to clarify the action of bound ligands; these ligands or chemicals are the drugs that could selectively inhibit the activity of the Cyclooxygenase-2 (COX-2) enzyme for inflammatory disease [41,42]. Therefore, COX-2 protein (PDB ID: 5KIR) was selected to execute a protein-ligand docking experiment in this study [41,43].

#### 2.9.3. Molecular Docking and Interactions Investigation

Protein-ligand docking is an appropriate technique to understand the protein-ligand bounded conformation and explains the molecular mechanism of small drug-like entities in cellular pathways [44]. The GC-MS-based fifty-two identified compounds that were used for molecular docking and a three-dimensional structure of the COX-2 enzyme (PBD ID: 5KIR) in PDB format were imported to the MOE software [45]. Heteroatoms, 3D protonation, and water molecules, along with the default ligand attached to the target protein, were removed to prepare the protein for the docking procedure. An active site was identified in the selected protein (5KIR) based on the previous literature [41], and structural optimization was performed by following parameters, such as the addition of hydrogen atoms and energy minimization with the Amber14 force field method was applied with chiral constraints and geometrical parameter. By using the surfaces and maps panel module, the transparency of the front and the back surface was adjusted and resulted in the information of significant residues in the selected substrate-binding pocket of 5KIR protein in native conformation [41]. MOE software creates a database of 52 compounds identified from experimental studies to perform molecular docking simulations and saves them with MDB extension for further analysis. Top-ranked poses were subjected to refinement and calculation of binding free energies (ΔG), which is evaluated by scoring function (GBVI/WSA dg) as described by Aldeghi et al. [46]. A reliable scoring scheme that results in the docking score of the correct binding poses was established by the number of molecular interactions (hydrogen, Pi, and Van der Waals interactions) documented in the literature of Ahmad et al. [47] and generated by Discovery Studio [43]; MOE database of the docked complexes was visualized carefully for understanding the mode of binding interactions of COX-2 inhibitors bound in the selected site of the target protein.

#### 2.9.4. Molecular Dynamics Simulations

Molecular dynamic (MD) simulations are a popular and interesting technique to understand the selected docked complex at a further atomic level, as reported by Bibi et al. [38]. For this study, the best-docked complex was selected for the MD simulation analysis at 100 ns; hence the conformational stability of the ligand bound in the vicinity of the active binding pocket of COX-2 protein in the system was analyzed for practical applications [48]. The study was divided into three major phases of MD simulation application. Initially, the parameter files were fixed, then subsequently moved towards the pre-processing, and finally performed the simulations as documented in the data of Ahmad et al. [47]. The antechamber module is important for preparing the files in AMBER20 software, as stated by Lee et al. [48]. Complex libraries and different MD simulations were fixed using ligand and protein information, and with the help of the Leap module, it was solvated at 12 Å, and appropriate measurement was accomplished. Molecular interacting residues were resolute with the help of the force field (ff14SB) [49]. The requirement of charge neutralization was fulfilled by the addition of Na+ ions to the system. In the system pre-processing, the selected binding energies were retrieved after the optimization of the system many times, hydrogen atoms were abated for the 500 steps, and the solvation energy was calculated for the 1000 steps by keeping the average limit of 200 kcal/mol-Å^2^, energy minimization was followed by the system carbon alpha atoms, complete set for once again until 1000 steps with the pragmatic scale of 5 kcal/mol-Å^2^, and similarly, for non-heavy atoms; 300 steps of minimization was performed with the scale of 100 kcal/mol-Å^2^. Hence, the system was heated for 300 K by NVT ensemble explained with Langevin dynamics and SHAKE algorithm as stated by Krautler et al. [50] and restrict the number of hydrogen bonds (HBs). The equilibration was attained by 100-ps. MD-simulation system was assisted with pressure using NPT ensemble parameters and fixed the system carbon alpha atoms. MD-simulation was subsequently performed using the time scale of 2-fs until 100 ns. By using CPPTRAJ, inter-and intra-molecular interactions were noted, keeping the cut-off range of distance at 8.0 Å [51]. For molecular and behavioral explanations, different trajectories were generated to explain the stability of the system with the help of the Visual molecular dynamic (VMD) tool, as earlier described by Humphrey et al. [52].

#### 2.9.5. Binding Free Energy (BFE) Estimation

Amber 20 was used for the estimation of binding interaction and the solvation-free energies for the COX-2 enzyme. Enzyme/protein-ligand complex was subjected to MM-PBSA calculation for absolute BFE estimation, which is the sum of gas-phase and solvation-free energies during MD simulation for 100 ns, as stated by Daina et al. [53], and its corresponding MM-GBSA calculations were performed with aims to develop the distinction between the bounded and detached presentation of solvated conformations of the selected potential target COX-2 as described by Sander et al. [54] Mathematically, the BFE can be analyzed by the Following Equation (3).
(3)$$\Delta G_{\mathrm{bind},\mathrm{solv}}=\Delta G_{\mathrm{bind},\mathrm{vacuum}}+\Delta G_{\mathrm{solv}},\;\mathrm{target}\;\mathrm{protien}-\mathrm{ligand}-\left(\Delta {G\;}_{\mathrm{solv}},\mathrm{ligand}+\Delta G_{\;\mathrm{solv},\;}\mathrm{target}\;\mathrm{protein}\right)$$

For all three conditions of the MM system, the solvation energy referred to as the transfer of molecules from the gas phase to solvent was assessed by resolving any one Poisson Boltzmann (PB) or Generalized Born (GB) equation. Therefore, it contributes to the electrostatic role of the solvation phase. Similarly, it permits the calculation of empirical terms for hydrophobic assistances, as presented in Equation (4).
(4)$$\Delta G_{\mathrm{solv}}=G_{\mathrm{electrostatic},\;\epsilon =80}-G_{\mathrm{electrostatic},\;\epsilon =1}+\Delta G_{\mathrm{hydrophobic}\;}$$

The estimation of the average interaction energy among the ligand and protein gives to delta-ΔG _vacuum_ (Equation (5)).
(5)$$\Delta G_{\mathrm{vacuum}}=\Delta E_{\mathrm{molecular}\;\mathrm{mechanics}}-T\cdot \Delta S$$

#### 2.9.6. In Silico Pharmacokinetic/ADMET Profile Calculations

Based on docking results, the best compound was used for the calculation of the ADMET (absorption, distribution, metabolism, excretion, and toxicity) profile, and it is a significant criterion for drug-like screening of chemical compounds, as stated by Khan et al. [55]. For ADMET profile estimation, SwissADME, as stated by Daina et al. [53], and Data-Warrior tools as reported by Sander et al. [54], were used.

### 2.10. Statistical Analysis

The data of the current study were taken in triplets and estimated using one-way analysis of variance (ANOVA), followed by Bonferroni’s test at significance level *p* = 0.05 represented (*) and 0.01 denoted as (**) using two-way ANOVA. While the Sidak’s multiple comparisons test [*p* = (ns > 0.9999, **** <0.0001)], for statistical authenticity. However, for antioxidant significance, a nonlinear regression graph was marked among % inhibition and concentration of the tested samples, and the IC_50_ was estimated via the GraphPad prism 9 programs for windows (GraphPad software, San Diego, CA, USA, 2020) by applying the below equation.
Y = 100/1 + (ˆHillSlope)
The equation can be explained as:
1 = Denote the concentration of the inhibitor.
Indicates the inhibitor’s reaction.
HillSlope indicates the steepness of the curve.

## 3. Result and Discussion

The current study was undertaken to evaluate the chemical ingredients and determine the microbial, free radicals scavenging, antinociceptive, and analgesic effects of EOs of *S. edelbergii* to scientifically validate their multiple therapeutic applications. The chemical constituents in plants are screened through various chromatographic techniques, including column chromatography, high-performance liquid chromatography, and gas chromatography-mass spectrometry.

### 3.1. GC-MS Analysis

The essential oils serve as alternative remedies aromatherapy contributed to their valuable capacities, the EOs are extracted from medicinal plants via hydro-distillation, microwave oven, and steam distillation, and the active ingredients are highlighted in GC-MS analysis. These bioactive chemical ingredients serve as a basis for the pharmaceutical industries to use the active chemical ingredients for their diverse biomedical applications as available drugs become ineffective over time. Plants are categorized as medicinal due to the presence of chemical constituents that are affected by various topographic, climatic, and numerous other factors that alter the composition of the bioactive ingredients. Fifty-two compounds were identified in the essential oil of the understudy plant *S. edelbergii* via GC-MS analysis, which contributed to around 89.52% (Table 1). The dominant compounds among the screened constituents were 3-Oxomanoyl oxide, followed by 24-Norursa-3,12-diene, methyl 7-abieten-18-oate, and β-eudesmol with 10.09%, 8.05%, 7.02%, and 6.39% respectively (Figure 1). These constituents were reported in some species, as described by Bekana et al. [56], and in the genus *Scutellaria,* earlier reported by Kurkcuoglu et al. [57] Furthermore, the essential oil of the under-study plant contains α-pinene (0.08%), myrcene (0.03%), and heptadecane (0.14%) in fewer amounts as compared to the same constituents, which were earlier reported from *S. diffusa* and *S. heterophylla* documented by Cicek et al. [58] at the quantity of 0.2%, 0.5%, and 0.2%, respectively. However, caryophyllene was observed in higher amounts (7.4%), and caryophyllene oxide (6.8%) from various species of the genus *Scutellaria* reflected in the literature of Formisano et al. [59], while the species of the same genus *S. edelbergii* depicted the same constituents in low amount with the amount of 2.30% and 3.94%, respectively. In addition to that, some species of the same genus *Scutellaria* possess the same constituents as reported in our studied plant as stated by Mamadalieva et al. [24].

These chemical ingredients as serve as antioxidants, anti-cancer, and antimicrobial agents, as reported by Hussain et al. [60] and Mahmood et al. [61] These constituents have antimicrobial activities, as stated by Utegenova et al. [27], as well as the capacity to scavenge free radicals, cure inflammation, and relieve pain, as reported by Surendran et al. [28]. The EOs present in the selected plant *S. edelbergii* were also present in some plants belonging to the same genus *Scutellaria* as reflected in the studies reported by Kasaian et al. [62], Yilmaz et al. [63], and lawson et al. [64].

### 3.2. Antibacterial Capacities

The SEEO were tested against the human pathogenic bacterial strains, *K. pneumonia*, *P. aeruginosa*, *E. coli*, and *E. faecalis*, for MIC, MBS, optical density (OD_600nm_), and inhibitory concentrations fifty (IC_50_) for 1/2MIC, MIC, and 2MIC concentrations displayed in Figure 2A,B, respectively. The zone of inhibitions was also determined at low to high doses, as well as compared with the standard (Levofloxacin) and negative control (DMSO) (Figure 3A). For SEEO at MIC, the IC_50_ for *K. pneumonia* was 84.60%, *P*. *aeruginosa* was 87.14%, *E. coli* was 87.96%, and *E. faecalis was* 83.70%. The current findings of SEEO exhibited considerable ability against the tested Gram-negative and Gram-positive bacterial strains. In addition, appreciable resistance of 15.8 ± 0.03 and 21.2 ± 0.02 mm was exhibited against the *E. faecalis* from low to high doses, as compared to levofloxacin 18.1 ± 0.01 and 23.7 ± 0.02; the *E. coli* and *P. aeruginosa* were also susceptible to SEEO. The essential oil was known for its antimicrobial significance, as stated by Seow et al. [65]. The substantial antibacterial activity of the SEEO might probably be due to caryophyllene [66], γ-eudesmole [67], linoleic acid, methyl ester [61], methyl stearate as stated by Adnan et al. [68], methyl pimar-8-en-18-oate [69], methyl-7-abieten-18-oate [70], 24-norursa-3,9(11),12-triene,24-norursa-3,12-diene [60], α-pinene and β-pinene [71], α-pellandrene [72], sabinene [27], psi-limonene [73], (+)-4-carene antibacterial resistance [74], α-campholenal, and L-pinocarveol can resist microbes [16], which was reported in the understudy plant (Table 1), represented the analysis depicts all these active fractions and compounds, which possess these antibacterial activities. However, Gram-positive strains are more susceptible to the tested sample as compared to Gram-negative bacterial strains, which agrees with the findings of Kasaian et al. [62], Skaltsa et al. [75], and Skaltsa et al. [76] in some species of *Scutellaria*. In addition that our findings were not agreed with the data reported by Bogdan et al. [77] for the essential oils of *Lavandula angustifolia* as the constituents in the plant might be influenced by phenological behaviors as stated by Moisa et al. [78] for *Thymus valgaris*.

### 3.3. Antifungal Significance

Similarly, the SEEO also produced a substantial effect against the tested fungal strains *C. albicans* and *F. oxysporum* in a dose-dependent manner. The MIC and MFC of the respective fungal strains, OD_600nm_ and IC_50_, are shown in Figure 2C and 2D, respectively. IC_50_ of SEEO at MIC was 93.93% and 77.54%, respectively, for *C. albicans* and *F. oxysporum*. Similarly, the maximum zone of inhibition (Figure 3B). It was observed against the SEEO displayed 14.3 ± 0.33 and 19.6 ± 0.05 mm resistance against *C. albicans* as compared with clotrimazole 17.6 ± 0.11 and 23.4 ± 0.4, respectively, from low to high doses. The antifungal effect is attributed to the existence of the bioactive compounds present in the SEEO, such as α-phellandrene [79], sabinene [27], α-phellandrene, dimer [79], thunbergen [80], thunbergol [81], and methyl dehydroabietic [82], which were stated in our GC-MS analysis (Table 1). The previous literature revealed that essential oils have a promising ability to overcome the complications caused by the fungus, as described by Singh [83]. Our findings were consistent with the research conducted by Yu et al. [84] and Zhu et al. [85] on the species of the same genus S. *barbata* and *S. strigillosa,* respectively. In addition, our outcomes also agreed with Zahra et al. [23] for the *S. multicaulis* and *S. bornmuelleri*. The current findings equated with the reported literature, using the same method and dose, as well due to similar genus. However, our data do not support the literature presented by Ullah et al. [86] for the essential oils of *Ochradenus arabicus* essential oils various parts due to the variation among the plant family and habitat, which influence the quality and quantity of the chemical ingredients.

### 3.4. Antioxidant Significance

The free radicals scavenging activity of different concentrations of SEEO was determined via DPPH and ABTS assays. The SEEO showed significant antioxidant activity due to the presence of diverse bioactive ingredients. The EOs of *S. edelbergii* presented a substantial mechanism to neutralize the free radicals in the DPPH assay with IC_50_ = 125.0 ± 0.19 µg/mL, while in the ABTS assay, the activity of the EOs was observed with an IC_50_ = 153.0 ± 0.31µg/mL). Furthermore, the antioxidant effect of the EOs of the studied plant was assessed to the ascorbic using the same concentrations and exhibited an IC_50_ = 70.19 ± 0.16 µg/mL and IC_50_ = 90.70 ± 0.32 µg/mL for the DPPH and ABTS bioassays, respectively (Figure 4). Our results strongly comply with those stated by Mamadalieva et al. [24] for some species of *Scutellaria*. The significance of scavenging the free radicals described by Lawson et al. [64] by the plant species belonging to the same genus *Scutellaria* consented to our data. However, different compounds were identified through GC-MS analysis in the same plant, *S. edelbergii* n-hexane crude oil reported by Shah et al. [31], as compared to the essential oil extracted through hydro-distillation. Furthermore, the essential oils extracted through hydro-distillation were observed to be efficient in neutralizing the free radicals as compared to the n-hexane-extracted crude oils of the same plant; hence, the mode of extraction also affects the quality and quantity of the compounds.

Our findings about the details of essential oils as a source of aromatherapy also conform to the data described by Mot. et al. [87] for *Salvia officinalis* essential oils and essential oils of *Ochradenus arabicus,* as stated by Ullah et al. [86] This highlights the idea that each molecule has a unique chemical structure that determines characteristic biochemical, physiological, pharmacological, toxic, etc. properties, and highlights this strong interdependence as reflected in the literature described by Glevitzky et al.’s [88] statistical analysis of the relationship between antioxidant activity and the structure of flavonoid compounds. This might be attributed to the difference in the chemical constituents among the essential and crude oils of the same plant. The main constituent responsible for scavenging the free radical is β-myrcene, linalool, α-campholenal, cuminal, and (-)-β-bourbonene, as revealed by Surendran et al. [28], Kamatou and Viljoen [89], Zhang et al. [79], and Ghasemi et al. [90]. It also complied with the literature by Khalilov et al. [91], with mainly flavonoids and phenols acting as promising antioxidant agents.

### 3.5. Analgesic Potential

The analgesic effect of SEEO was determined using Swiss albino mice at low to high doses (25, 50, and 100 mg/kg) and showed significant pain-relieving activity when tested against pain induced by acetic acid, as shown in Table 2. The tested sample at a dose of 25 mg/kg depicted 33.33% and exhibited significant inhibition of 43.11% and 54.71% at a concentration of (50 and 100 mg/kg). Moreover, the normal saline has no effect, and the standard (aspirin) presented 64.49% in writhes reduction tested in the Swiss albino mice. The current findings were consistent with the literature presented by Uritu et al. [92] for the *Scutellaria* species, as well as by Mondal et al. [35], which described the analgesic significance in the leaves of *Eucalyptus camaldulensis* and with the study presented by Chen et al. [93], Sarmento-Neto et al. [15], and Mishra et al. [94], in which the EOs were extracted using the same hydro-distillation methods. By applying different oils extraction methods, the same plant yields different types of compounds; thus, the essential oils extracted via hydro-distillation were much more effective in curing pain as compared to the crude oils obtained through column chromatography of the n-hexane fraction as a part of the continuous studies earlier documented by Shah et al. [31] The promising analgesic activity was depicted by the SEEO due to the presence of bioactive ingredients such as β-myrcene previously described by Surendran et al. [28]; γ-eudesmole, as reflected in the literature of Aati et al. [29]; tau-muurolol, documented by Hameed et al. [95]; and some other bioactive compounds with the capacity to reduce pain. Furthermore, our data also matched with the findings of Liang et al. [96]; the root EOs of *Illicium lanceolatum and B. persicum* EOs, as described by Hajhashemi et al. [97], presented the significant potential to cure pain. Moreover, *S. rufinervis* is an aromatic plant comprising Eos, which also has aromatherapeutic significance, especially in relieving pain, as documented by Santos et al. [98].

### 3.6. Anti-Inflammatory Capabilities

The SEEO were tested to cure the inflammation induced by carrageenan in the experimental animals, as displayed in Table 3. The SEEO demonstrated considerable capability with 53.10% inhibition at the dose of 25 mg/kg, whereas the tested samples at the concentrations of 50 mg/kg and 100 mg/kg displayed 57.93% and 64.13% inhibition, as compared to Diclofenac Sodium, which depicted 71.72% inhibition. Moreover, the normal saline does not affect the paw diameter of the swiss albino mice caused by carrageenan. The use of essential oil for remedial practices is documented in the various literature and specially to reduce paw edema, as stated by Colares et al. [99] The potential exhibited by the SEEO to cure inflammation is also supported by the literature given by Mogosan et al. [100] in some *Mentha* species and further validated by Boukhatem et al. [101] in some other species of the same family Lamiaceae. Moreover, the SEEO extracted through hydro-distillation offered a significant ability to cure inflammation when equated with the crude oils extracted from the same plant *S. edelbergii* using column chromatography of the n-hexane fraction described by Shah et al. [31] However, the available literature stated that the bioactive compounds responsible for curing inflammation are β-myrcene as conveyed by Surendran et al. [28], sabinene described by Zhang et al. [79], linalool illustrated by [89], myrtenal documented by Dragomanova et al. [102], and many other potent chemical ingredients with the capability to cure the inflammation and act as an anti-inflammatory agent. Our data were consistent with the finding noticed for *Jatropha curcas* described by Adesosun et al. [103] However, our data are not equated with the reported literature by Apel et al. [104] for *Myrciaria tenella* and *Calycorectes sellowianus* essential oils; they presented high significance as compared to our studied plant.

### 3.7. Molecular Docking and Interactions Analysis

The structural information of the COX-2 enzyme (PDB ID: 5KIR) was used. The prepared biomolecule structure without any bounded ligand and considerable active binding residues pocket for the protein-ligand interaction analysis was performed by Sander et al. [54]. A database of 52 bioactive compounds was identified through GC-MS and subjected to MOE with MDB extension and performed protein-ligand docking simulations by the Dock module of MOE software [54]. Phytochemicals with the best binding poses and molecular interactions with the significant amino acids intricate the mechanism of inhibition of COX-2 enzyme to manage inflammatory disease. By the evaluation of the selected active binding site residues, targeted docking was applied. Dock score and RMSD values calculated for the database of 52 compounds with COX-2 protein by MOE software are enlisted in Table 4. While the selected highest dock scored four compounds were demonstrated with the best binding poses, noteworthy binding interactions with active site residues of COX-2 enzyme within the range of 4.5 Å are presented in Figure 5, and a summary of docking results is enlisted in Table 4.

All four selected potential hits presented the Van der Waals, carbon-hydrogen, alkyl, and Pi-alkyl bonding, and most of the selected active site is hydrophobic (Table 5). The dock score range is from −7.9497 to −7.4221 kcal/mol and shows that these hits were bounded in the best conformation within the target COX-2 protein selected active site (Figure 5).

#### 3.7.1. Molecular Dynamic Simulation Analysis

Based on the most stable and best-scored conformation, the compound (**47**) bounded with COX-2 enzyme complex was subjected to MD simulation analysis for 100 ns and retrieved very good results. The superimposed complex at 0 ns and 100 ns is demonstrated in Figure 6. The calculated RMSD value of the superimposed complex is 1.157 Å, and it is a very good value means there is minor fluctuation observed during the MD simulation system at 100 ns; hence, to differentiate the superimposed structures, it is demonstrated in different colors, such as protein at 0 ns is presented in Cyan and ligand in yellow color, while protein at 100 ns is presented in Sienna and ligand in coral color. Critical structural changes are shown in the zoomed view in Figure 6.

During the MD simulation run, 2D plots were generated to explain the fluctuating behavior of the docked complex at different time frames during MD simulation productions, as explained in many previous studies for the identification of COX-2 inhibitors, as described by Razzaghi et al. [105] These plots are important for the MD simulation’s statistical analysis; hence it could be significant to decode the backbone stability and flexibility of the residues during the different time frames of MD simulations as carried out by Razzaghi et al. [105] RMSD was calculated as the small atoms convergence from a reference state (Figure 7A), while the RMSF of a protein explained the protein’s dynamic nature that contributed to the system’s overall versatility and residual mobility from its mean position (Figure 7B). The average RMSD value was noted as 2.722 Å, and the average RMSF value was noted as 1.351 Å for the COX-2 inhibitor, Methyl 7-abieten-18-oate complex; the results perceived that those key amino acids residues were correlated in the binding interactions at 0 ns and 100 ns, showing maximum correlation with the docking results, and minor fluctuations of the COX-2 macromolecules residues were observed from the initial state during MD simulation runs during 0–100 ns simulation period. The radius of gyration (Rg) was evaluated to confirm the distribution of protein elements and define the equilibrium conformation of the system during MD simulations; it is expected that the high and low values of radius of gyration describe the magnitude of the system in terms of molecules tight packing scheme as reflected in the literature of Lobanov et al. [106]. The estimated average Rg of the system was 24.338 Å (Figure 7C). Beta-factor (BF) of the protein complex is a very complicated and challenging aspect; it explains the thermal residual deviation throughout the MD simulation run (Figure 7D) and seems effectively correlated with RMSF, and hereafter approves the stability of the system. BF and RMSF are complements of each other and explain the overall simulation system stability along with the aspects of elements flexibility of protein as stated by Raniolo and Limongelli [107].The average BF of the system analyzed for COX-2 inhibitor and enzyme complex is 79.941 Å.

The frequency of the HBs plays an important function in the overall stability of the protein-ligand complex and increases the capacity of HBs, which upsurges the binding attributes of the biomolecule Alkorta et al. [108]. These HBs were obtained through by VMD hydrogen bond plugin. The estimated maximum number of HBs between COX-2 residues and inhibitor atom was 9, and the average number of HBs was 1.779, respectively, during MD analysis until 0–100 ns (Figure 8A). Solvent accessible surface area (SASA) is a very expedient examination in which changes in the accessibility of the COX-2 protein to solvent were determined. The stability of each complex was perceived during the simulation, and the average value of SASA was calculated at 145.520 nm^2^ (Figure 8B).

#### 3.7.2. Binding Free Energies (BFE) Calculations (MM-GBSA/MM-PBSA)

In molecular mechanics (MM), PBSA and GBSA energy model generation are the acceptable and necessary techniques used in the biomolecular investigation of BFE calculations of COX_2 and inhibitor complex and support the analysis of protein folding and stability in drug design and discovery Ismail et al. [109]. The MM energy of the selected complex (ΔTOTAL) followed by generalized surface area Born (MM/GBSA) outlines the well-organized estimations deprived of any loss in terms of accuracy and Poisson-Boltzmann (MM/PBSA) benchmark parameters (Genheden and Ryde, 2015), could be significant to expose the promising macromolecule COX-2—methyl 7-abieten-18-oate ligand complex in pure water. The total energy estimated for the MM/GBSA and MM/PBSA model of the Methyl 7-abieten-18-oate complex is −50.9841 kcal and −42.4442 kcal/mol, respectively. For molecular mechanics’ energy parameters, more influence was observed from the gas-phase energy (ΔG gas) when assessed to extremely irrelevant influences from the solvation energy (ΔG solv). In the MM/GBSA model of Methyl 7-abieten-18-oate-complex, the ΔG gas energy was estimated as −92.2470 kcal/mol; however, in the MM/PBSA model with minor difference energy parameters, estimation was recorded as −92.2092 kcal/mol. For the Methyl 7-abieten-18-oate-complex, the ΔG solv energy for the MM/GBSA model was projected as 40.6321 kcal/mol, though in the case of MM/PBSA, 50.4040 kcal/mol was noticed. For Methyl 7-abieten-18-oate-complex, the electrostatic forces play a role in the stability of the complex, and the system estimated by the MM forces field in PBSA is in the range of total premeditated energy calculated as −32.3777 kcal/mol for Methyl 7-abieten-18-oate-complex; for GBSA models, it is −32.1854 kcal/mol. Similarly, the Van der Waals forces are also estimated from MM associated with the system stability as −60.8915 kcal/mol and −60.2943 kcal/mol. For the Methyl 7-abieten-18-oate-complex complex, the electrostatic energy impact (EGB and EPB) to the ΔG solv was observed, and the principal constraint following towards the out-of-the-range calculations in MM/GBSA solv energy. The surface area energy highlighted as ESURF computed in the MM/GBSA model is −6.0429 kcal/mol. In the MM/PBSA model, the two important ENPOLAR and EDISPER, the repulsive and attractive free energies presented as −4.6961 kcal/mol and Zero kcal/mol. For Methyl 7-abieten-18-oate-complex complex, the electrostatic energy impact, EGB, and EPB to the ΔG solv were counted as non-favorable results in MM/GBSA solv energy. The individual total bounded conformation presenting the free energy for the selected COX-2 macromolecule as receptor and selected phytochemicals complex as a ligand (47) are explained in Table 6.

#### 3.7.3. In Silico Pharmacokinetic/ADMET Profile Calculations

Many studies explained the importance of pharmacokinetic/ADMET profile estimation as carried out by Chandrasekaran et al. [110] for the screening of databases to identify potential drug-like and lead-like compounds that could be better tolerable in the discovery and development of novel drug candidates for the management of inflammatory diseases, as reflected in the literature by Sudha et al. [111], ADMET properties are calculated by SwissADME, as described by Daina et al. [53], and Data-warrior tools, as performed by Sander et al. [54] Physicochemical properties of the selected compound, such as molecular weight, partition coefficient/lipophilic parameters (logP values), hydrogen bond acceptor, hydrogen bond donor, total polar surface area, molar refractivity, and rotatable bond are important drug-like characteristics calculated for the selected compound as documented by Chandrasekaran et al. [110]. In the initial phase, drug discovery protocols promote the calculation of drug-likeness, water-solubility, pharmacokinetics, and toxicity estimations, along with the medicinal chemistry perspective, as mentioned for selected compounds in Table 7. The enlisted chemical descriptors of the selected compound are in the acceptable range without any violation, as described by Lipinski [112] and Veber et al. [113] theory of drug-likeness. Lipophilicity and water solubility classes also presented very good outcomes. Gastrointestinal drug absorption as stated by Kimura and Higaki, [114] and blood-brain barrier permeability [115] are also in the range of acceptable pharmacokinetic parameters. Compound **47** showed CYP2C19 and CYP2C9 inhibitory potential and presented nonsubstrate characteristics for not P-glycoprotein. Prediction of Log Kp value (skin permeation), as reported by Alonso et al. [116], is good for selected compound −4.15 cm/s, PAINS alert and Brenk alert, supported by the medicinal chemistry parameter evaluation, as reported by Bibi et al. [117]. Our results enlisted the minor violations as Brenk alert; one isolated alkene entity is required to be optimized. It is suggested to improve before moving a drug to the next phase of development. Synthetically compound **47** is highly accessible with a score of 4.69, and in silico toxicity estimations with four major aspects of mutagenicity, tumorigenicity, irritant, and reproductive effects are completed in an acceptable range. Therefore, this selected lead compound has presented very good ADMET results, as given in Table 7.

## 4. Conclusions

Due to their diverse health-promoting benefits, Physico-chemical, biological properties, and versatile chemical structures, essential oils are currently gaining considerable scientific attention as alternative effective and safe therapeutic candidates for a variety of disorders. The SEEO are reported for the first time for in vitro and in vivo pharmacological activities, and they showed significant antimicrobial activity, the ability to scavenge free radicals, a significant effect on pain, and the potential to cure inflammation when compared to their standards. The results indicate that the essential oils of *S. edelbergii* contain a bioactive chemical constituent, Methyl 7-abieten-18-oate (**47**), which has the potential to act as a possible candidate molecule against microbes, as an antioxidant, and as an effective pain reliever and anti-inflammatory. Computational analysis has further validated the anti-inflammatory potential of methyl 7-abieten-18-oate to suppress COX-2 enzyme activity. COX-2 protein and methyl 7-abieten-18-oate docked results were very good in terms of docked score (−7.8704 kcal/mol) and binding interactions with the functional residues; furthermore, when subjected to MD simulation (for 100 ns), they presented correlation with docking results with minor fluctuations. ADMET characteristics additionally supported that compound **47** could be recommended for further investigations in a clinical test; it could be a future drug for the management of inflammation, which is the cause of other metabolic disorders, so it is highly required to be controlled. Furthermore, more research is needed to identify the accountable constituents with the capacity to carry out the observed activities.

## Figures and Tables

**Figure 1 antioxidants-11-01723-f001:**
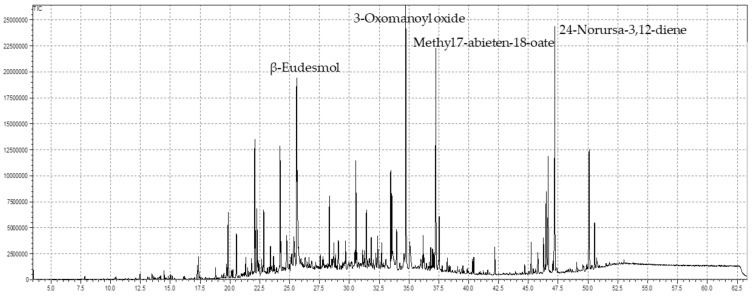
GC chromatogram of SEEO.

**Figure 2 antioxidants-11-01723-f002:**
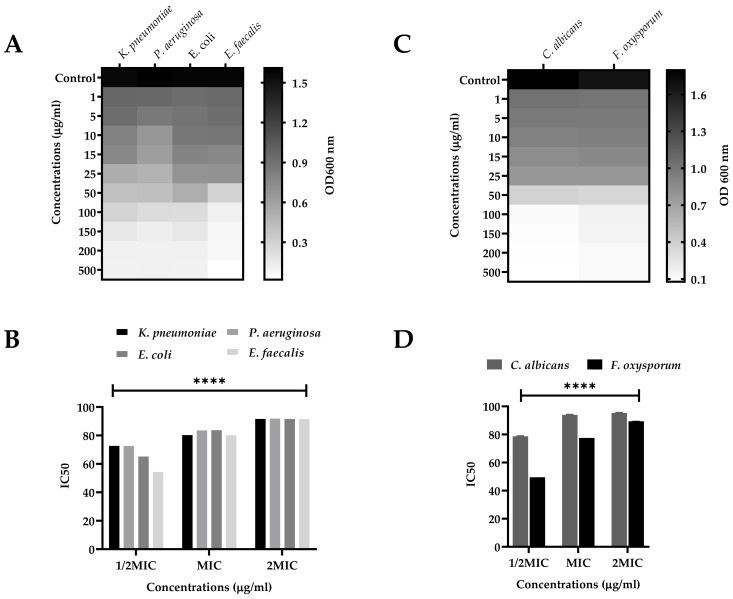
(**A**,**B**) antibacterial OD_600nm_ and IC_50_ of *Klebsiella pneumoniae, Pseudomonas aeruginosa*, *Escherichia coli,* and *Enterococcus faecalis* against various concentrations of essential oil of *S. edelbergii;* (**C**,**D**) Antifungal OD_600nm_ and IC_50_ of the potential of essential oils of *S. edelbergii* against *Candida albicans* and *Fusarium oxysporum* against various concentrations. Data were taken in triplicate (*n* = 3) and analyzed through two-way ANOVA, via Tukey’s multiple comparison test, *p* = **** < 0.0001).

**Figure 3 antioxidants-11-01723-f003:**
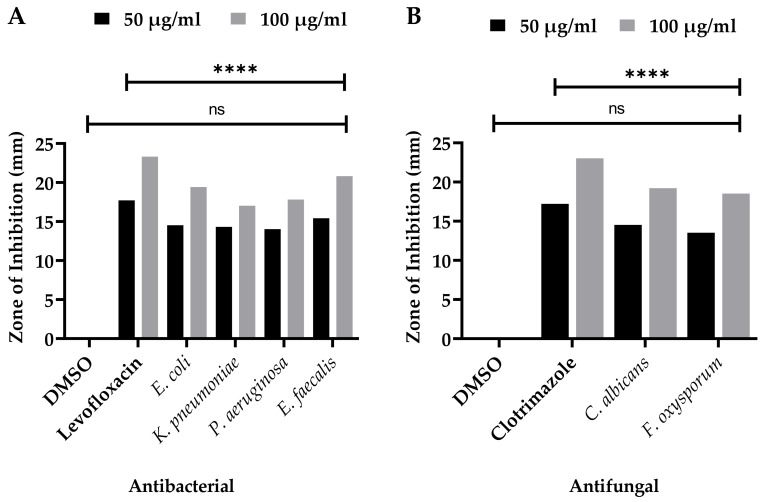
Zone of inhibition’s (**A**) antibacterial and (**B**) antifungal potential zone of inhibitions of essential oils of *S. edelbergii*; DMSO = negative control; levofloxacin and clotrimazole = positive control. Data were taken in triplicate (*n* = 3) and analyzed through two-way ANOVA, via Sidak’s multiple comparison test, ns = >0.9999, *p* = **** < 0.0001).

**Figure 4 antioxidants-11-01723-f004:**
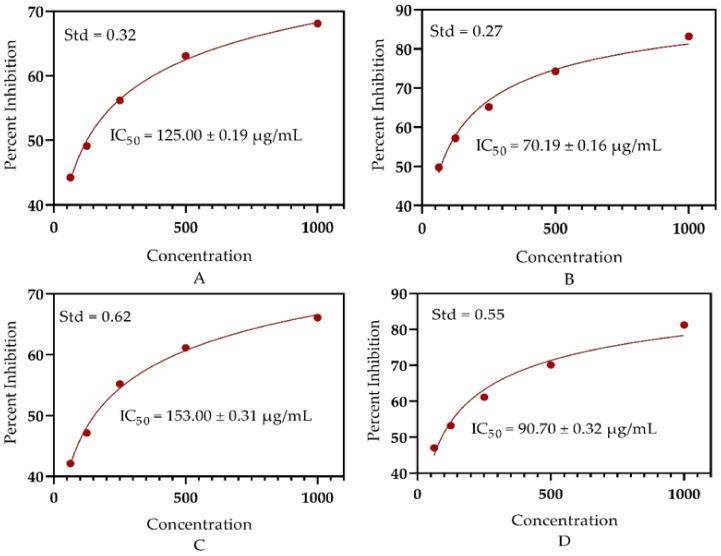
(**A**) antioxidant potential of the *S. edelbergii* essential via DPPH assay, (**B**) antioxidant activity of ascorbic acid at DPPH assay, (**C**) antioxidant activity of SEEO via ABTS assay, (**D**) free radicals scavenging effect of ascorbic acid via ABTS assay, (Std) standard deviation.

**Figure 5 antioxidants-11-01723-f005:**
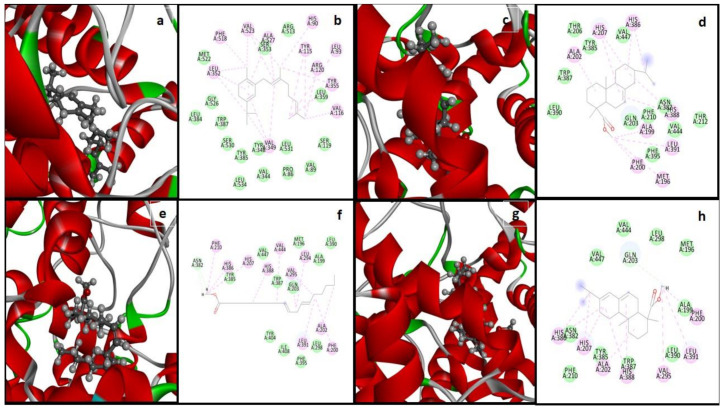
Four compounds (((**41**) **a**,**b**), ((**47**) **c**,**d**), ((**44**) **e**,**f**), and ((**49**) **g**,**h**)) with the highest dock score represent the best binding poses within the active binding site of Cyclooxygenase-2 (COX-2) protein (PBD ID: 5KIR). Three-dimensional view of the best-bounded conformation of docked ligands (**a**,**c**,**e**,**g**) and a two-dimensional plot showing significant protein-ligand binding interactions (green colored circles present Van der Waals interactions and carbon-hydrogen bond, while purple-colored circles present alkyl and Pi-alkyl interactions) (**b**,**d**,**f**,**g**).

**Figure 6 antioxidants-11-01723-f006:**
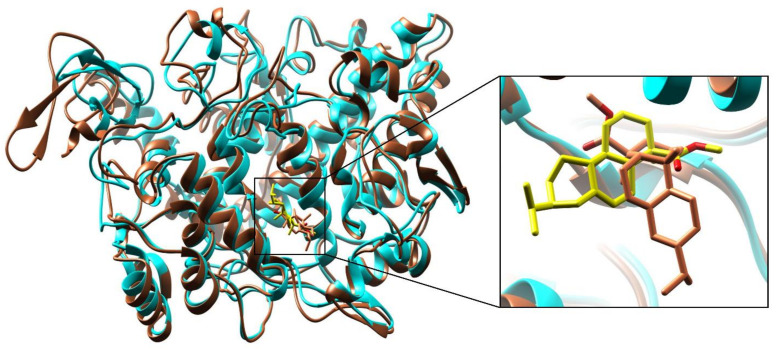
Superimposition of the compound (**47**) bounded with Cyclooxygenase-2 (COX-2) at 0 ns and 100 ns. The calculated RMSD value is 1.157 Å; protein at 0 ns is presented in Cyan and ligand in yellow color, while protein at 100 ns is presented in Sienna and ligand in coral color. Critical structural changes are shown in zoomed view.

**Figure 7 antioxidants-11-01723-f007:**
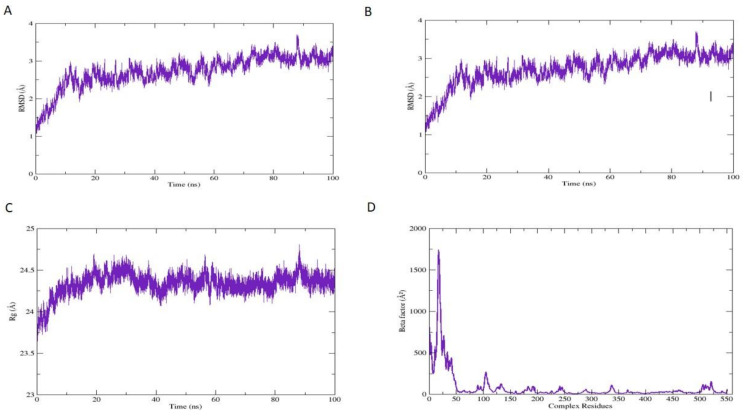
Root means square deviation (RMSD) of COX-2 protein and Methyl 7-abieten-18-oate complex (**A**) in-unit Angstrom (Å) is shown at Y-axis, while the variation in bonded conformation through time in nanoseconds (ns) is shown at X-axis. Root Mean Square Fluctuation (RMSF) of protein COX-2 (**B**) in-unit Angstrom (Å) is shown at the Y-axis, while the X-axis shows the index of the residue during 0–100 ns. The radius of gyration (Rg) of protein and COX-2 inhibitor complex (**C**) is shown at Y-axis in-unit Angstrom (Å), while X-axis shows the variation in bonded conformation throughout simulations. A beta factor of protein COX-2 (**D**) is shown at Y-axis in-unit Angstrom (Å), while X-axis shows the index of the residue until 100 ns.

**Figure 8 antioxidants-11-01723-f008:**
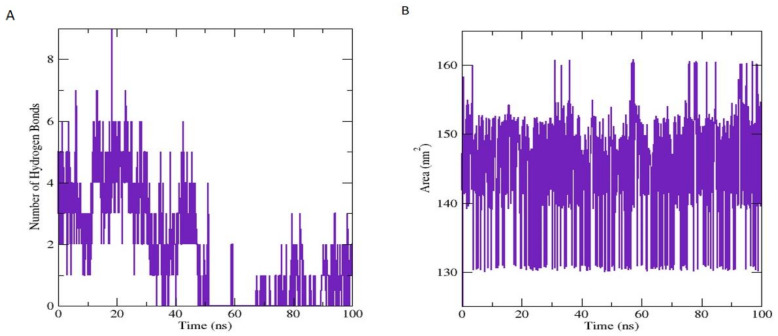
The number of hydrogen bonds of COX-2 inhibitor (Methyl 7-abieten-18-oate) complex until 0–100 ns (**A**). Y-axis shows the calculated number of hydrogen bonds, while X-axis shows variation in timespan during molecular dynamic simulations. While solvent-accessible surface area (SASA) of COX-2 protein (**B**). Y-axis shows the thermodynamic calculations of changes in COX-2 inhibitor complex surface area in unit nanometers square (nm^2^) (Å), while X-axis shows the residue’s stability time during 0–100 ns.

**Table 1 antioxidants-11-01723-t001:** GC-MS detected EOs of *S. edelbergii*.

C. No	Name of the Compounds	Rt	Contents (%)	RI_Rep._	RI_calc._
**1**	α-Pinene	7.83	0.08	931	942
**2**	β-Pinene	8.91	0.02	943	982
**3**	β-Myrcene	9.37	0.03	958	999
**4**	α-Pellandrene	9.75	0.04	964	1013
**5**	Sabinene	10.43	0.10	969	1021
**6**	psi-Limonene	10.48	0.02	992	998
**7**	(+)-4-Carene	11.24	0.03	1017	1026
**8**	Linalool	12.31	0.02	1081	1104
**9**	Nonanal	12.46	0.14	1080	1110
**10**	α-Campholenal	13.10	0.08	1102	1134
**11**	L-Pinocarveol	13.47	0.15	1143	1147
**12**	cis-Verbenol	13.61	0.09	1131	1153
**13**	α-Phellandren-8-ol	14.19	0.09	1148	1174
**14**	Terpinen-4-ol	14.48	0.24	1160	1185
**15**	α-Terpineol	14.82	0.10	1172	1198
**16**	Myrtenal	15.01	0.13	1175	1205
**17**	Cuminal	16.11	0.08	1214	1248
**18**	Carvone	16.21	0.08	1229	1251
**19**	Bornyl acetate	17.26	0.59	1273	1292
**20**	Phenol,2-methyl-5-(1methylethyl)-	17.36	0.60	1278	1296
**21**	α-Terpinyl acetate	18.79	0.62	1322	1354
**22**	(-)-β-Bourbonene	19.71	0.46	1386	1392
**23**	β-Elemene	19.84	2.09	1398	1401
**24**	Caryophyllene	20.54	2.30	1421	1428
**25**	Humulene	21.32	0.40	1454	1462
**26**	gamma-muurolene	21.79	0.46	1471	1483
**27**	β-eudesmene	22.07	4.10	1478	1494
**28**	α-Selinene	22.26	1.83	1500	1503
**29**	Cadina-1(10), 4-diene	22.83	1.90	1514	1529
**30**	Elemol	23.38	0.93	1535	1554
**31**	Caryophyllene oxide	24.20	3.94	1575	1591
**32**	γ-Eudesmole	24.793	0.54	1627	1619
**33**	tau-Cadinol	25.35	0.60	1628	1646
**34**	β-Eudesmol	25.58	6.39	1644	1658
**35**	tau-Muurolol	25.63	4.43	1628	1660
**36**	Ar-Turmerone	25.77	0.19	1638	1667
**37**	α-Phellandrene, dimer	28.33	1.52	1801	1811
**38**	Linalyl phenylacetate	29.08	1.51	1945	1953
**39**	Thunbergen	29.70	1.74	1934	1942
**40**	p-Camphorene	30.47	2.47	1977	1986
**41**	Geranyl.alpha-terpinene	31.10	1.72	1962	1973
**42**	Thunbergol	31.83	0.50	2073	2082
**43**	Verticiol	32.38	0.96	2106	2118
**44**	Linoleic acid, methyl ester	33.46	4.12	2071	2080
**45**	3-Oxomanoyl oxide	33.96	10.09	2133	2140
**46**	Methyl pimar-8-en-18-oate	36.97	1.40	2231	2297
**47**	Methyl 7-abieten-18-oate	37.25	7.02	2164	2178
**48**	Methyl dehydroabietate	37.53	2.79	2293	2335
**49**	Methyl abietate	38.20	1.36	2339	2379
**50**	24-Norursa-3,9(11),12-triene	46.50	3.70	3042	3057
**51**	24-Norursa-3,12-diene	47.21	8.05	3105	3062
**52**	24-Norursa-3,12-dien-11-one	50.09	6.68	3351	3308
	Identified compounds		89.52		

Identified compounds: Elution order on HP-5MS column. RI_rep._: Retention Index from the database (NIST, 2011). C. No = Compound number; RI_calc._ = Retention Index calculated_._

**Table 2 antioxidants-11-01723-t002:** Analgesic significance of essential oils of *S. edelbergii*.

Treatment	Dose Conc.	No. of Writhes Mean ± SEM	% Reduction in Writhes after 45 min
Acetic acid	1 mL	27.6 ± 0.03	
Normal saline	1 mL	27.3 ± 0.05	-
Aspirin	1 mL	9.8 ± 0.02	64.49
SEEO	25 (mg/kg)	18.4 ± 0.02 **	33.33
	50	15.7 ± 0.03 **	43.11
	100	12.5 ± 0.05 **	54.71

(Acetic acid) Inducer, (Normal saline) negative control, (Aspirin) standard, (SEEO) *S. edelbergii* essential oil, *p* = 0.01 denoted as (**).

**Table 3 antioxidants-11-01723-t003:** Anti-inflammatory significance of *S. edelbergii* EOs.

		Changes in Paw Diameter in Swiss Albino Mice (Mean ± SEM)				
Samples Used	Dose Conc.	1 h	2 h	3 h	Av. Paw Diameter	% Inhibition
Carrageenan	1 mL	1.21 ± 0.03	1.42 ± 0.03	1.74 ± 0.05	1.45 ± 0.03	
NS	1 mL	1.18 ± 0.02	1.40 ± 0.04	1.71 ± 0.02	1.43 ± 0.03	-
Standard	50 (mg/kg)	0.49 ± 0.02	0.42 ± 0.03	0.33 ± 0.01	0.41 ± 0.02	71.72
SEEO	25	0.73 ± 0.04	0.68 ± 0.02	0.63 ± 0.03	0.68 ± 0.03 *	53.10
	50	0.67 ± 0.01	0.62 ± 0.03	0.56 ± 0.02	0.61 ± 0.04 *	57.93
	100	0.58 ± 0.06	0.53 ± 0.04	0.46 ± 0.02	0.52 ± 0.03 *	64.13

(NS) normal saline, (standard) diclofenac Na, (SEEO) *S. edelbergii* essential oils, *p* = 0.05 denoted as (*).

**Table 4 antioxidants-11-01723-t004:** Dock score and RMSD values were calculated for 52 compounds with COX-2 protein.

Numbering	Score	RMSD	Numbering	Score	RMSD
**1**	−5.323	1.2178	**27**	−6.234	0.651
**2**	−5.392	1.44	**28**	−6.198	3.501
**3**	−5.421	1.0412	**29**	−6.144	2.123
**4**	−5.422	1.042	**30**	−6.543	1.577
**5**	−5.364	1.4674	**31**	−5.922	2.437
**6**	−5.319	1.3645	**32**	−6.477	0.716
**7**	−5.311	0.6295	**33**	−6.054	1.429
**8**	−5.691	1.9427	**34**	−6.032	1.647
**9**	−5.758	1.0724	**35**	−6.503	1.481
**10**	−5.550	1.1354	**36**	−6.248	1.524
**11**	−5.677	1.1992	**37**	−7.130	2.209
**12**	−5.566	0.8491	**38**	−7.043	2.447
**13**	−5.490	0.5436	**39**	−6.715	0.929
**14**	−5.564	2.2633	**40**	−6.711	1.964
**15**	−5.536	0.4417	**41**	−7.949	0.639
**16**	−5.565	0.4645	**42**	−6.640	2.445
**17**	−5.160	1.2301	**43**	−6.237	2.991
**18**	−5.529	0.7864	**44**	−7.626	1.845
**19**	−6.139	2.7774	**45**	−6.808	1.376
**20**	−5.323	0.8127	**46**	−6.731	1.114
**21**	−6.572	1.2823	**47**	−7.870	1.920
**22**	−6.329	2.1625	**48**	−6.802	2.784
**23**	−6.006	1.2073	**49**	−7.422	2.663
**24**	−5.9901	1.3014	**50**	−6.3748	4.1046
**25**	−5.9146	2.6874	**51**	−6.315	2.3077
**26**	−6.3482	0.9651	**52**	−6.6796	3.2421

**Table 5 antioxidants-11-01723-t005:** Summary of molecular docking results of top 4 dock score compounds with the COX-2 target protein.

Numbering	Dock Score (kcal/mol)	Functional Residues	Binding Interactions
**41**	−7.9497	PRO86 (A), VAL89 (A), HIS90 (A), LEU93 (A), TYR115 (A), VAL116 (A), SER119 (A), ARG120 (A), VAL344 (A), TYR348 (A), LEU352 (A), SER353 (A), TYR355 (A), LEU359 (A), LEU384 (A), TYR385 (A), TRP387 (A), ARG513 (A), PHE518 (A), MET522 (A), VAL523 (A), VAL523 (A), GLY526 (A), ALA527 (A), SER530 (A), LEU531 (A), LEU534 (A).	Van der Waals, Carbon-hydrogen bond, Alkyl, Pi-Alkyl
**47**	−7.8704	MET196 (A), ALA199 (A), PHE200 (A), ALA202 (A), GLN203 (A), THR206 (A), HIS207 (A), PHE210 (A), THR212 (A), ASN382 (A), TYR385 (A), HIS386 (A), TRP387 (A), HIS388 (A), LEU390 (A), LEU391 (A), PHE395 (A), VAL444 (A), VAL447 (A).	Van der Waals, Carbon-hydrogen bond, Alkyl, Pi-Alkyl
**44**	−7.6261	MET196 (A), ALA199 (A), PHE200 (A), ALA202 (A), GLN203 (A), HIS207 (A), PHE210 (A), LEU294 (A), VAL295 (A), LEU298 (A), LEU390 (A), LEU291 (A), ASN382 (A), TYR385 (A), HIS386 (A), TRP387 (A), HIS388 (A), LEU390 (A), PHE395 (A), TYR404 (A), ILE408 (A), VAL444 (A), VAL447 (A).	Van der Waals, Carbon-hydrogen bond, Alkyl, Pi-Alkyl
**49**	−7.4221	MET196 (A), ALA199 (A), PHE200 (A), ALA202 (A), GLN203 (A), HIS207 (A), PHE210 (A), VAL295 (A), LEU298 (A), ASN382 (A), TYR385 (A), HIS386 (A), TRP387 (A), HIS388 (A), LEU390 (A), LEU391(A), VAL444 (A), VAL447 (A).	Van der Waals, Carbon-hydrogen bond, Alkyl, Pi-Alkyl

**Table 6 antioxidants-11-01723-t006:** Binding free energies of the COX-2 protein and selected potential bioactive compound complex.

MM/GBSA Model				MM/PBSA Model			
Energy (E) Component	Average Values	Standard Deviation Values	Standard Error of Mean Values	Energy (E) Components	Average Values	Standard Deviation Values	Standard Error of Mean Values
Van der Waals	−60.8915	2.3527	0.2469	Van der Waals	−60.2943	4.3561	0.4356
EEL	−32.3777	5.9753	0.7345	EEL	−32.1854	5.9826	0.7432
EGB	46.3402	4.5936	0.5086	EPB	55.1001	5.1322	0.4423
ESURF	−6.0429	0.1611	0.0163	ENPOLAR	−4.6961	0.1102	0.0110
				EDISPER	0	0	0
ΔG gas	−92.2470	6.5432	0.6949	ΔG gas	−92.2092	6.3912	0.6391
ΔG solv	40.6321	4.6615	0.4661	ΔG solv	50.4040	5.0137	0.5014
Δtotal	−50.9841	4.1324	0.4319	ΔTOTAL	−42.4442	5.2303	0.5140

**Table 7 antioxidants-11-01723-t007:** Summary of in silico ADMET profile estimated for selected one compound.

Descriptors	Methyl 7-Abieten-18-Oate (47)
Formula	C_21_H_34_O_2_
Molecular weight	318.49 g/mol
Number of rotatable bonds	3
Number of hydrogen bond acceptors	2
Number of hydrogen bond donors	0
Molar refractivity	97.01
Total polar surface area	26.30 Å^2^
Lipophilicity (Log P)	4.71
Water Solubility (Log S)	−4.46
Solubility Class	Moderately soluble
**Drug-pharmacokinetic parameters**	
Gastrointestinal absorption	Yes
Blood-brain barrier permeability	Yes
P-glycoprotein substrate	No
CYP1A2 inhibitor	No
CYP2C19 inhibitor	Yes
CYP2C9 inhibitor	Yes
CYP2D6 inhibitor	No
CYP3A4 inhibitor	No
Log Kp (skin permeation)	−4.15 cm/s
**Drug-like characteristics**	
Lipinski rule	Acceptable
Veber rule	Acceptable
Drug-likeness	Yes
Drug-likeness score	0.55
**Parameters of medicinal chemistry**	
PAINS alert	None
Brenk alert	1 alert: isolated alkene
Synthetic accessibility score	4.69
Toxicity	None

## Data Availability

The data presented in this study are available in the article.

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
