# Peer review of "Bridging the Chemical Profile and Biomedical Effects of Scutellaria edelbergii Essential Oils"

_antioxidants, 2022, doi:10.3390/antiox11091723_

Round 1

Reviewer 1 Report

The authors have extensively studied the plant Scutellaria edelbergii (Molecules. 2021 Jun 19;26(12):3740; Molecules. 2021 Dec 18;26(24):7676; Molecules. 2022 Feb 16;27(4):1322.) On this occasion they present the analysis of the essential oil, considering the chemical characterization and biological properties (antimicrobial, antioxidant, anti-inflammatory, analgesic) and computational analysis.

In the introduction a number of ideas are presented but the relationship between them is not shown. -Antimicrobials-resistance to antibiotics, oxidations, antioxidants, inflammation, etc. I suggest integrating the information and presenting the importance and novelty of the study; it is only mentioned in one sentence on line 71: Oxidation, inflammation and pain are interconnected disorders that affect each other... antimicrobials???

In the introduction a number of ideas are presented but the relationship between them is not shown.

Antimicrobials-resistance to antibiotics, oxidations, antioxidants, inflammation, etc.

I suggest integrating the information and presenting the importance and novelty of the study; it is only mentioned in one sentence on line 71: Oxidation, inflammation and pain are interconnected disorders that affect each other

Line 84-85, the authors present natural remedies with few side effects, however, there are many plants and even essential oils that are toxic.

Line 97, include the species descriptor.

The introduction is a paragraph of more than one page, please use punctuation marks to separate the paragraphs, main ideas, etc.

Extraction of essential oil:

How much plant did they use?

What part of the plant was used? What do you mean by specimen?

Why did the authors dry the plant to obtain the essential oil? , they are commonly extracted from the fresh plant to contain the volatiles. When the plant is dried, these components can be lost

How long did the hydro-distillation take?

How many oils? One or more? Why do they refer to them in plural SEEOs

Authors are advised to consult Adams, R. P. (2007). Identification of essential oil components by gas chromatography/mass spectrometry, 4th Edition. Allured Publ., Carol Stream

What was the yield of oil extraction? Did you calculate the density?

K. pneumoniae, P. aeruginosa, E. coli, and E. faecalis were characterized by Chairman Dr. Hazir Rahman, Department of Microbiology AWKUM, Mardan. What characteristics do the strains have? Where were they isolated from? Or where did they come from?

What is the positive and negative control?

Number of repetitions? In tests with bacteria?

For fungi, they also calculated the minimum bactericidal concentration (MBC)??? Line 179

For Fusarium oxysporum they used the same method as for Candida. The growth of a mycelial fungus is different from that of a yeast

Fusarium oxysporum is a phytopathogen, why use it in this study?

What was the molar concentration of DPPH?

For antioxidant activity it is important to calculate the IC 50 or CA50, Number of repetitions?

In trial 2.7. Estimation of Analgesic Activity

Review equation 2

A= Indicates the paw edema induced through carrageenan????? line 228

It is suggested to do a cox-2 inhibition assay to check the results in silico

How do the authors know that all 52 components are active? Did you check the activity of each one in the techniques proposed in this study?

GC-MS analysis presented that the SEEOs have 52 bioactive compounds… line- 376

The low concentration of α -pinene, β-pinene, and β-myrcene can be explained by the use of dried plant to obtain the essential oils.

What quantities of these monoterpenes have been reported in other species of the genus?

For the identification of the compounds, did you compare the mass spectra?

The discussion about the antimicrobial activity of the oils is not clear. The authors attribute the activity to many components of the oil that are present in trace amounts. It is suggested that they focus on the most abundant.

Figure 3. Missing units on the Y axis of panel B and D

The positive controls levofloxacin and clotrimazolewere evaluated in each strain? They always presented the same zones of inhibition?

Do the results of the IC50 of ascorbic acid agree with the literature?

In the graphs include the standard deviation

The hydrodistillation method is commonly used to obtain essential oils, studies of other species (Eucalyptus) based solely on the extraction method cannot be compared. Authors are suggested to focus on the chemical composition of the oil.

The statement in line 483 is more than obvious:

The same plant by applying different oils extraction methods yields different types of compounds,

The references that are mentioned in the MS discussion, studied the biological effect of the pure compound (analgesic, anti-inflammatory, antioxidant, etc)? It is not valid to refer to other species that produce essential oils with similar activities, since mixtures are studied; unless the chemical composition of the species in the literature, eg Jatropa, is very similar to that of S. edelbergii.

What do you mean by the concentration of 1 ml of acetic acid? one molar? 1mL?

Why the authors do not refer to aromatherapy in the discussion if it is a central part of the research because it is in the title of the MS

Why is compound 47, Methyl 7-abieten-18-oate, not discussed further?

Before reaching clinical trials of compound 47, its ability to inhibit cox in vitro must be tested. There are several colorimetric tests to do this.

Author Response

Response to the Reviewer 1st

Reviewer Comment: The authors have extensively studied the plant Scutellaria edelbergii (Molecules. 2021 Jun 19;26(12):3740; Molecules. 2021 Dec 18;26(24):7676; Molecules. 2022 Feb 16;27(4):1322.) On this occasion they present the analysis of the essential oil, considering the chemical characterization and biological properties (antimicrobial, antioxidant, anti-inflammatory, analgesic) and computational analysis.

Author response: Worthy reviewer, thanks for your valuable suggestions that will help to improve our article. Yes, we selected the under-study plant because of its multiple health benefits as recommended by the local inhabitants, studied its various aspect and continued so the article in hand is one of the aspects of the continuous project which will lead us to the isolate the responsible compounds from the essential oils and used them for their multiple biomedical applications.

Reviewer Comment: In the introduction, several ideas are presented but the relationship between them is not shown. -Antimicrobials-resistance to antibiotics, oxidations, antioxidants, inflammation, etc. I suggest integrating the information and presenting the importance and novelty of the study; it is only mentioned in one sentence on line 71: Oxidation, inflammation, and pain are interconnected disorders that affect each other... antimicrobials???

Author response: As earlier mentioned it is a part of the continuous project based on its local practices so here as the previous studies are comparing its significance and promising source based on which the promising source will be proceeded for its pharmacological importance in the next step. So numerous ideas were discussed. However, the suggested part of the article was revised and highlighted in the revised version.

Reviewer Comment: In the introduction, several ideas are presented but the relationship between them is not shown. Antimicrobials-resistance to antibiotics, oxidations, antioxidants, inflammation, etc.

Author response: Worthy reviewer the revised version has been made according to your recommendation and highlighted.

Reviewer Comment: I suggest integrating the information and presenting the importance and novelty of the study; it is only mentioned in one sentence on line 71: Oxidation, inflammation, and pain are interconnected disorders that affect each other.

Author response: Worthy the main ideas were separated accordingly in the revised version and your suggestion has been addressed accordingly in the revised version. As here in hand manuscript, the selected plant has been used earlier for the mentioned studies and presented therapeutic significance so here to continue and highlight its significance the essential oils were used for the first time from the selected plant.

Reviewer Comment: Line 84-85, the authors present natural remedies with few side effects, however, there are many plants and even essential oils that are toxic.

Author Response: Yes worthy reviewer, some of the toxicological effects of the essential oils added and highlighted in the revised version.

Reviewer Comment: Line 97, include the species descriptor.

Author response: Worthy reviewer, added and highlighted in the revised version.

Reviewer Comment: The introduction is a paragraph of more than one page, please use punctuation marks to separate the paragraphs, main ideas, etc.

Author Response: Worthy reviewer, the paragraphs were separated accordingly

Reviewer Comment: Extraction of essential oil: How much plant did they use?

Author Response: The suggested details were added and highlighted in the methodology section of the manuscript.

Reviewer Comment: What part of the plant was used? What do you mean by specimen?

Author response: Worthy reviewer as the selected plant is herbaceous by habit so whole plant was used. The typos specimen replaced by the species in the revised version. However, Specimen is those plant taxa which are collected, identified, tagged, preserved and deposited in herbarium which can be further used for future studies as a reference species.

Reviewer Comment: Why did the authors dry the plant to obtain the essential oil? they are commonly extracted from the fresh plant to contain the volatiles. When the plant is dried, these components can be lost

Author response: Dear Reviewer, you are right, as we were planning to send it to another country (University of Nizwa, Oman) from Pakistan for GC-MS analysis and it was not possible to shift the fresh plant. That’s why we dried the plant under shade to avoid the loss of volatiles and other type of contamination.    

Reviewer Comment: How long did the hydro-distillation take?

Author response: The hydro distillation time was 6 hrs. three times for the extraction of oils. 

Reviewer Comment: How many oils? One or more? Why do they refer to them in plural SEEOs.

Author response: It is only one oil but due to typos mistake we wrote SEEOS and replaced  with SEEO in the revised article.

Authors are advised to consult Adams, R. P. (2007). Identification of essential oil components by gas chromatography/mass spectrometry, 4th Edition. Allured Publ., Carol Stream.

Response: We have cited the reference in the revised version of the manuscript.

Reviewer Comment: What was the yield of oil extraction? Did you calculate the density?

Response: The yield of oil extraction was added in the revision (highlighted). Currently we did not calculate the density but we shall keep in mind to calculate in the future.

Reviewer Comment: K. pneumoniae, P. aeruginosa, E. coli, and E. faecalis were characterized by Chairman Dr. Hazir Rahman, Department of Microbiology AWKUM, Mardan. What characteristics do the strains have? Where were they isolated from? Or where did they come from?

Author response: Worthy reviewer, all the tested samples were clinical isolate identified by Dr. Hazir Rahman, a well-known Associate Professor of Microbiology, who have an established lab in the Department of Microbiology, Abdul Wali Khan University, Mardan. That’s why we acknowledged his attribution in this regard.

The microbes have the following features.

  1. pneumoniae is pathogenic to the respiratory tract, it is mucoid on a culture plate and aerobic, gram-negative bacilli, capsulated, and lactose fermenter. It is pathogenic in the urinary tract and respiratory tract.
  2. aeruginosa is a highly pathogenic antimicrobial resistant bacteria with green pigmentation feature on culture agar and facultative aerobic ability. It is a rod-shaped gram-negative bacterium. It is known for a broad spectrum of clinical symptoms in immunocompromised patients including nosocomial infections.
  3. coli are gram-negative, motile, with large, thick, white, moist colonies. Causes intestinal and extraintestinal tract infections.
  4. faecalis is group D streptococcus, gram-positive cocci, in pairs or chains, with no capsules and motility. Mainly found in wound infections, urinary and biliary tract infections.

Reviewer comment: What is the positive and negative control?

Author response: Thanks for your valuable comments positive control was levofloxacin for antibacterial and clotrimazole was used for the antifungal assay. However, DMSO was used as a negative control.

Reviewer comment: The number of repetitions? In tests with bacteria?

Author response: For scientific validity and authenticity, the entire data were taken in triplicates and listed as (Mean ± SEM).

Reviewer comment: For fungi, they also calculated the minimum bactericidal concentration (MBC)??? Line 179

Author response: Thanks for your valuable attention toward a typo mistake, indeed, it is minimum fungicidal concentrations (MFC), we did rectify the mistake in the main manuscript.

Reviewer comment: For Fusarium oxysporum they used the same method as for Candida. The growth of a mycelial fungus is different from that of yeast.

Author response: Thanks for your valuable comments. Both fungal strains can be grown on potato dextrose agar, for more reference, the papers have been attached. 

References: 1. https://doi.org/10.2991/ahsr.k.200523.002,

  1. DOI: 10.12692/ijb/7.6.74-91,

3.https://doi.org/10.3390/antiox11081446,

  1. https://doi.org/10.3390/molecules26247676).

Reviewer Comment: Fusarium oxysporum is a phytopathogen, why use it in this study?.

Author response: Thanks for your valuable and very technical question, Yes, it’s a phytopathogen, abundant in the soil, air, and on plants. Fusarium species can be pathogens in humans who are immunocompromised and have rarely been reported to cause disease in immunocompetent individuals also in humans causes keratitis (Zapater, 1986) endophthalmitis (Tiribelli et al., 2002) onychomycosis (Gianni et al., 1997 and Romano et al.,1998) disseminated in cutaneous infections and many other human health concerned disorders so that is why screened the plant against to highlight its significance.  It may cause mycotoxicosis in humans following ingestion of food that has been colonized by the fungal organism. In humans, it can also cause disease that is localized, focally invasive, or disseminated. So, that’s why we choose this strain in our experiment. For reference (DOI: 10.1097/00001432-200004000-00005).

Reviewer comment: What was the molar concentration of DPPH?

Author response: Worthy reviewer, the concentration used for DPPH is 76 micromolar or (0.076 mM).

Reviewer comment: For antioxidant activity, it is important to calculate the IC 50 or CA50, the Number of repetitions.

Author Response: To determine the free radical significance the most cited approach IC 50 is used. However, CA50 is a tumor marker and is used to diagnose gastrointestinal malignancies. All the data of the invitro and invivo were taken in triplicates.

Reviewer comment: In trial 2.7. Estimation of Analgesic Activity

Author response: Corrected in the revised version.

Reviewer comment: Review equation 2. A= Indicates the paw edema induced through carrageenan ????? line 228.

Author response. Worthy reviewer, thanks for your suggestion, reviewed and the line was confusing so elaborated properly.

Reviewer comment: It is suggested to do a cox-2 inhibition assay to check the results in silico. How do the authors know that all 52 components are active? Did you check the activity of each one in the techniques proposed in this study?

Author response: Thanks for your valuable comments. The computational screening will give us a clue of the most active compounds. Furthermore,  literature has been studied and every compound in the literature was reported for its substantial biological activities. However, as it is a part of our continuous project and we have a plan to isolate some of the chemical ingredients for the proposed studies.

Reviewer comment: GC-MS analysis presented that the SEEOs have 52 bioactive compounds… line- 376.

Author response: Corrected and highlighted in the revised version.

Reviewer Comment: The low concentration of α -pinene, β-pinene, and β-myrcene can be explained by the use of the dried plant to obtain the essential oils.

Author response: Yes, worthy reviewer you are right the essential oils are mostly extracted from the fresh samples but as I mentioned in the earlier response that we have 

to send the sample to another country University of Nizwa Sultanate Oman so was not possible for the plant species to be sent in fresh form so that is why we dried the sample which affects the quantity but the essential oil can be also extracted from dry samples as well DOI: 10.3109/13880209.2010.491083 for Senecio rufinervis D.C. Essential Oil and many other plants essential oil are reported from dry samples.

Reviewer Comment: What quantities of these monoterpenes have been reported in other species of the genus?

Author response: We have added the reported monoterpenes along with their quantity from the same genus Scutellaria compared. The revised part was highlighted in the results section of the GC-MS analysis.

Reviewer Comment: For the identification of the compounds, did you compare the mass spectra?

Author response: No, we have identified the compounds through reported RI. We have added the RI (reported) values in table 1 in the revised version.

Reviewer comment: The discussion about the antimicrobial activity of the oils is not clear. The authors attribute the activity to many components of the oil that are present in trace amounts. It is suggested that they focus on the most abundant.

Author response: Worthy reviewer, the suggested section has been properly revised and highlighted in the revised version and the data of compound present in maximum amount were also inserted.

Reviewer comment: Figure 3. Missing units on the Y axis of panels B and D

Author response: Worthy reviewer, inserted as per your suggestion.

Reviewer comment: The positive controls levofloxacin and clotrimazole were evaluated in each strain? They always presented the same zones of inhibition.

Author response: Worthy reviewer, the positive controls levofloxacin was run for antibacterial while clotrimazole for antifungal was evaluated and presented maximum significance as compared to the essential oil of the understudy plant presented in the result section of the article through graphical representations.

Reviewer comment: Do the results of the IC50 of ascorbic acid agree with the literature?

Author response: Yes. the actual data is very rare to match but we found similarity in the IC50 of ascorbic acid very closely reflected in the literature with references https://doi.org/10.3390/antiox11050808 and DOI: 10.3390/molecules27165197

Reviewer comment: In the graphs include the standard deviation.

Author response: Dear reviewer, the Std values were added in the revised graphs.

Reviewer comment: The hydro-distillation method is commonly used to obtain essential oils, studies of other species (Eucalyptus) based solely on the extraction method cannot be compared. Authors are suggested to focus on the chemical composition of the oil.

Reviewer comment: The statement in line 483 is more than obvious: The same plant by applying different oils extraction methods yields different types of compounds.

Author response: worthy reviewer, the suggested line has been rephrased and highlighted in the revised version.

Reviewer comment: The references that are mentioned in the MS discussion, studied the biological effect of the pure compound (analgesic, anti-inflammatory, antioxidant, etc). It is not valid to refer to other species that produce essential oils with similar activities since mixtures are studied; unless the chemical composition of the species in the literature, eg Jatropa, is very similar to that of S. edelbergii.

Author response: Thanks for your valuable suggestions. All the results and discussion sections were supplemented were additional data as per your recommendation and highlighted in the revised version of the manuscript.

Reviewer comment: What do you mean by the concentration of 1 ml of acetic acid? one molar? 1mL?

Author response: The I ml or 1mL is used for a 1-milliliter concentration of acetic acid.

Reviewer comment: Why the authors do not refer to aromatherapy in the discussion if it is a central part of the research because it is in the title of the MS.

Author response: The author express thanks for your valuable ideas. Various data on the essential oils extracted from the medicinal plant to highlight their significance has been added and highlighted in the revised manuscript.

Reviewer comment: Why is compound 47, Methyl 7-abieten-18-oate, not discussed further?

Author response: Our ongoing project is to isolate compounds. The computational screening shows that the mentioned compound has responsible for the therapy of inflammation so we will isolate and will screen for anti-inflammatory and other biological activities. As this study was to highlight the active ingredient so that is why is not discussed further.

Reviewer comment: Before reaching clinical trials of compound 47, its ability to inhibit cox in vitro must be tested. There are several colorimetric tests to do this.

Author response: Worthy reviewer, we have tested the essential oil which was a mixture of many compounds through computational analysis the compound is highlighted. Once we isolate we will proceed further first with in vitro and then in vivo screening to determine their ability against inflammation.

Reviewer 2 Report

The present manuscript is dedicated to the study of the essential oil of the medicinal plant Scutellaria edelbergii growing in Pakistan. The chemical composition was analyzed by GC-MS, antimicrobial, antiradical and anti-inflammatory properties of the oil were studied. The chemical and biological data were subjected to computational analysis, using molecular docking and interactions analysis and in silico pharmacokinetic /ADMET profile calculations. One of the major constituents of the essential oil, methyl 7-abieten-18-oate was found to have the potential of a possible candidate molecule against microbes, as an antioxidant, and as an effective pain reliever and anti-inflammatory agent. The methods applied are up-to-date and appropriate, the conclusions are supported by the results. There are only a few points which need additional attention by the Authors:

1.       The Authors claim that they have identified 52 bioactive compounds but this is the total number of identified compounds, are all of them bioactive.

2.       In Table 1, the Authors have to indicate which compounds were identified by comparison with authentic samples, and which by comparison with library RI and spectra. Also, they should include the experimental RI in this Table.

Author Response

Reviewer 2nd

Comments and Suggestions for Authors

Reviewer Comment: The present manuscript is dedicated to the study of the essential oil of the medicinal plant Scutellaria edelbergii growing in Pakistan. The chemical composition was analyzed by GC-MS,  antimicrobial, antiradical, and anti-inflammatory properties of the oil were studied. The chemical and biological data were subjected to computational analysis, using molecular docking and interactions analysis and in silico pharmacokinetic /ADMET profile calculations. One of the major constituents of the essential oil, methyl 7-abieten-18-oate was found to have the potential of a possible candidate molecule against microbes, as an antioxidant, and as an effective pain reliever and anti-inflammatory agent. The methods applied are up-to-date and appropriate, and the conclusions are supported by the results. There are only a few points that need additional attention by the Authors:

Author response: Thanks worthy reviewer for considering our article and appreciating our work. your suggestions and recommendation will help in the improvement of our article.

      Reviewer Comment: 1. The Authors claim that they have identified 52 bioactive compounds but this is the total number of identified compounds, are all of them bioactive.

Author response: Worthy reviewer, the compounds were identified in the mixture of the essential oil of the understudy sample however the literature reflects that each of the identified compounds has biological significance so that is why the term bioactive was used for all the identified compounds.

      Reviewer Comment: 2. In Table 1, the Authors have to indicate which compounds were identified by comparison with authentic samples, and which by comparison with library RI and spectra. Also, they should include the experimental RI in this Table.

Author response:  All the compounds were identified by comparing them with the reported RI values. We have included the experimental RI values in the table and highlighted them in the revised version of the manuscript.

Reviewer 3 Report

The paper Bridging the Chemical Profile and Biomedical Effects of Scutellaria edelbergii Essential Oils: A New Approach Toward Aromatherapy is interesting and could be consider for publication in Antioxidants. The paper is well documentedHowever, some corrections and improvements should be made:

it should be checked and corrected if uppercase or lowercase should be applied (for instance line 106, 196, 353, 390, 502, 506), also for name of standards, i.e. levofloxacin etc.

name of Van der Waals should be corrected, also kcal

equation - line 195 - should be corrected - also information regarding evaluation of studied samples (not only standards) should be added

line 209 temperature ±20oC - is it correct?

line 221 - what does it means: B.W. doses?

line 322 - should be protein

line 421, 426, Fig. 3  - should be: C. albicans, F. oxysporum

line 585 - citation should be corrected

Moreover, in some references issue number is in brackets, sometimes not

Author Response

Reviewer 3rd

Comments and Suggestions for Authors

Reviewer Comment: The paper Bridging the Chemical Profile and Biomedical Effects of Scutellaria edelbergii Essential Oils: A New Approach Toward Aromatherapy is interesting and could be considered for publication in Antioxidants.

Author response: Thanks worthy reviewer for considering our work for publication in the prestigious journal.

Reviewer Comment: The paper is well documented However, some corrections and improvements should be made and should be checked and corrected if uppercase or lowercase should be applied (for instance lines 106, 196, 353, 390, 502, 506), also for the name of standards, i.e., levofloxacin, etc.

Author response:  Worthy reviewer for appreciating our work. All the corrections you suggested were properly addressed accordingly and highlighted in the revised version

Reviewer Comment: name of Van der Waals should be corrected, also kcal

Author response:  Worthy reviewer, corrected as per your suggestion.

Reviewer Comment: Equation - line 195 - should be corrected - also information regarding the evaluation of studied samples (not only standards) should be added

line 209 temperature ±20oC - is it correct?

Author response:  Worthy reviewer, thanks for your valuable comments. The lines and corrections you recommended have been addressed accordingly in the revised version of the manuscript.

Reviewer Comment: line 221 - what does it mean: B.W. doses?

Author response:  Worthy reviewer, (B.W) denoted body weight which was earlier mentioned in the same assay and abbreviated as highlighted now so therefore I used the symbol here.

Reviewer Comment: line 322 - should be protein

Author response: corrected in the revision

Reviewer Comment: line 421, 426, Fig. 3  - should be: C. albicans, F. oxysporum

Author response: Corrected in the revised version in text and figure.

Reviewer Comment: line 585 - citation should be corrected

Moreover, in some references issue number is in brackets, sometimes not

Author response: Dear reviewer, the citation was corrected and all the references were made in uniform format according to the journal style in the revised version of the manuscript.

Round 2

Reviewer 1 Report

The authors made corrections to the manuscript

Reviewer 3 Report

Almost all the suggested corrections have been implemented. I would suggest a slight change in the formula notation:

% Scavenging activity = [(A − B) ⁄A] × 100